# Metabolic, Oxidative and Psychological Stress as Mediators of the Effect of COVID-19 on Male Infertility: A Literature Review

**DOI:** 10.3390/ijerph19095277

**Published:** 2022-04-26

**Authors:** Gesthimani Mintziori, Leonidas H. Duntas, Stavroula Veneti, Dimitrios G. Goulis

**Affiliations:** 1Unit of Reproductive Endocrinology, 1st Department of Obstetrics and Gynecology, Aristotle University of Thessaloniki Medical School, Papageorgiou Gneral Hospital, 56429 Thessaloniki, Greece; stavroulav2@hotmail.com (S.V.); dgg@auth.gr (D.G.G.); 2Unit of Endocrinology, Diabetes and Metabolism, Thyroid Section, Evgenideion Hospital, University of Athens, 20 Papadiamantopoulou Str., 11528 Athens, Greece; leonidas@duntas.gr

**Keywords:** COVID-19, male infertility, metabolic stress, oxidative stress, psychological stress

## Abstract

Over 300 million patients with coronavirus disease 2019 (COVID-19) have been reported worldwide since the outbreak of the pandemic in Wuhan, Hubei Province, China. COVID-19 is induced by the acute respiratory syndrome coronavirus 2 (SARS-CoV-2). The effect of SARS-CoV-2 infection on the male reproductive system is unclear. The aim of this review is to assess the effect of SARS-CoV-2 infection on male fertility and the impact of possible mediators, such as metabolic, oxidative and psychological stress. SARS-CoV-2 infection aggravates metabolic stress and directly or indirectly affects male fertility by reducing seminal health. In addition, SARS-CoV-2 infection leads to excessive production of reactive oxygen species (ROS) and increased psychological distress. These data suggest that SARS-CoV-2 infection reduces male fertility, possibly by means of metabolic, oxidative and psychological stress. Therefore, among other consequences, the possibility of COVID-19-induced male infertility should not be neglected.

## 1. Introduction

Since the outbreak of the coronavirus disease 2019 (COVID-19) pandemic in Wuhan, Hubei Province, China, over 300 million cases have been reported worldwide [1,2]. Severe acute respiratory syndrome coronavirus 2 (SARS-CoV-2), the causative agent of COVID-19, is an RNA virus and a member of the beta-coronavirus family consisting of 29,891 nucleotides and 9860 amino acids [3]. The virus, which has a diameter of 60–140 nm, acquired this name from its crown-like shape [4]. Virus transmission among humans occurs either through direct contact or with airborne transmission through aerosols. The most common ways of spread are coughing, sneezing, or oral, nasal or eye membrane contact [5]. The most common clinical presentation of COVID-19 includes fatigue, cough and fever. Less often, patients complain of headache, dizziness or gastrointestinal symptoms such as nausea, vomiting and abdominal pain. Other symptoms are anosmia and ageusia [6]. Age, obesity, comorbidities and indicators of organ dysfunction are risk factors for increased COVID-19 severity [7]. Furthermore, severe disease and fatality are more prevalent in males compared with females [8,9,10]. Consequently, the reproductive hormones have been implicated in SARS-CoV-2 infection pathophysiology.

Semen quality deteriorates immediately after the SARS-CoV-2 infection. These effects on semen quality may be the direct effect of the virus on semen or an indirect effect of the cytokine storm, which is induced by the infection and the increased oxidative stress [11].

All types of stress (metabolic, oxidative, psychological) affect male fertility adversely [12,13]. Infertility is defined as failure to achieve a clinical pregnancy after 12 months of unprotected sexual intercourse and affects 8–12% of couples worldwide. It is estimated that males are solely responsible for 20–30% of infertility cases, and they contribute to 50% of all cases. There are several causes of infertility that affect both sexes, such as hypogonadotrophic hypogonadism, hyperprolactinemia, infections, systemic diseases, cystic fibrosis and, in some cases, lifestyle factors. However, some causes of infertility affect males only, such as testicular deficiencies and semen deterioration due to aging and endocrine disruptors [14]. To understand the pathophysiology of male infertility, knowledge of the testicular structure is necessary. The testes are oval-shaped structures, covered by the tunica vaginalis, the outer layer, and the tunica albuginea, the inner layer. The testes consist of lobules that contain seminiferous tubules. The tubular epithelium consists of spermatogenic and Sertoli cells; the Leydig cells are located in the interstitial space among the seminiferous tubules [15]. Vascular endothelial and perivascular cells, as well as immune cells (mostly macrophages), are also present in the interstitial space [16]. Sperm is released by the Sertoli cells into the lumen of the seminiferous tubules for transport towards the epididymides. The latter, located on the posterior testicular margin, are connected to the vas deferens [17]. The epididymides are responsible for sperm maturation. Following this process, sperm is forwarded towards the seminal vesicles, a pair of glands behind the urinary bladder which secrete alkaline fluid that partly composes the semen. The prostate gland also produces fluid that, combined with the output of the seminal vesicles and the sperm, forms the semen [18]. Male infertility and male functional (non-classical) hypogonadism are often diagnosed during the investigation of couple infertility; they have been associated with oxidative and psychological stress as well as metabolic disorders (metabolic stress) [19].

Based on the hypothesis that SARS-CoV-2 infection impairs male fertility by increasing oxidative, metabolic and psychological stress, this manuscript aims to assess the impact of COVID-19 on male infertility (via the effects on testosterone secretion or semen parameters) and study the contribution of various types of stress as possible mediators (Figure 1).

## 2. SARS-CoV-2 and Male Infertility

### 2.1. SARS-CoV-2 and Testosterone

Although androgens, such as testosterone (T) and dihydrotestosterone (DHT), are produced in both sexes, their concentrations in males are approximately ten times those of females. The testes and the adrenal cortex secrete T, an anabolic hormone. After the age of 40 years, T serum concentrations decrease approximately 2% per year, leading to a 9.5% increase in hypogonadism in men over 40 years of age [20]. Νormal T production is inextricably linked to normal hypothalamic–pituitary–gonadal axis function. SARS-CoV-2 crosses the blood–brain barrier and results in neuroinflammation through the infected angiotensin-converting enzyme 2 (ACE2)–expressing cells. Hormonal imbalance could manifest due to this inflammation [21]. Male hypogonadism has been correlated with poor prognosis and mortality in patients with COVID-19 admitted to intensive care units [22]. Low T concentrations in men represent a risk factor for high morbidity and mortality [23]. More specifically, among 89 patients with COVID-19, 30 patients with non-COVID-19 respiratory tract infections and 143 age-matched controls, serum T concentrations were 186, 289 and 332 ng/dL, respectively (median values, *p* < 0.0001) [21]. Recently, a comparison between 286 patients with COVID-19 and 281 healthy controls showed lower T concentrations in the former (*p* < 0.0001) [24]. Another study reported that T concentrations fluctuated between patients with COVID-19 of varying severity: patients with more severe disease showed lower T concentrations compared with patients with disease of mild-to-moderate severity [median (range) 85.1 (0.21–532) vs. 315 (0.88–486) ng/dL, *p* < 0.001, respectively). When the T concentrations of patients with COVID-19 who were admitted to the intensive care unit were compared with those who did not need admission, the results were similar [median (range) 64.0 (0.21–337) vs. 286 (0.88–532) ng/dL, *p* < 0.001, respectively). Differences in T concentrations were also demonstrated between patients with COVID-19 who died and survivors [median (range) 82.9 (2.63–165) vs. 166 (0.21–532) ng/dL, *p* < 0.001, respectively) [25]. Another study confirmed the above results, where patients with severe COVID-19 had lower T concentrations (1.4 ng/mL) compared with patients with mild COVID-19 (3.5 ng/mL) (*p* = 0.005) [26]. While T concentrations are lower in males with more severe COVID-19 disease than those with milder disease, there are no differences in women with severe and mild COVID-19 disease [27]. Among hospitalized men with COVID-19, those with T concentrations <100 ng/dL had higher mortality risk [odds ratio (OR) 18.2, 95% confidence interval (CI) 2.3–144.6, *p* = 0.006] than those with T concentrations >230 ng/dL [28]. It is of note that T exerts biological actions other than those of reproduction, playing, inter alia, an anti-inflammatory role. Obesity and low T concentrations could trigger the cytokine storm that leads to the deterioration of COVID-19 patients’ clinical conditions [29]. Moreover, it is speculated that SARS-CoV-2 infection per se could rapidly cause depletion of androgenic action, resulting in severe or even fatal disease [30]. Further studies are needed to clarify whether T replacement therapy (TRT) could be beneficial in severely hypogonadal men with COVID-19 [31]. Furthermore, the SARS-CoV-2 spike protein exacerbates endothelial injury when the virus interacts with DHT [32,33].

### 2.2. SARS-CoV-2 and Semen

More than 20 viruses affecting fertility have been found in human semen [34]. However, as SARS-CoV-2 was not detected in prostatic secretions, semen or testes, virus transmission via the genital tract appears unlikely [35,36]. Prospective observational studies that assessed the semen 6–75 days after COVID-19 diagnosis failed to detect SARS-CoV-2 [37]. A study of 18 patients with COVID-19 demonstrated no presence of SARS-CoV-2 in semen during the acute or convalescent phase of COVID-19 [38]. Another study of 11 consecutive men with mean age 29.7 ± 4.5 recovering from COVID-19 proved that there was no presence of SARS-CoV-2 in semen by using reverse transcription polymerase chain reaction (RT-PCR). The samples were collected 19–59 days (median 44 days) after the positive test for SARS-CoV-2. Patients declared that there was no history of epididymo-orchitis or sexual dysfunction at the time of inclusion [39]. In 70 semen samples of patients with COVID-19, there was no detection of the viral nucleic acid; however, there was a deterioration in total sperm count and motility compared with healthy controls. Moreover, there was an association between recovery duration and semen parameters: the longer the recovery period, the worse the sperm quality [40]. Contradicting these results, one study reported several cases in which SARS-CoV-2 was detected in semen during the acute and the recovery phase of COVID-19 [41], implying potential sexual transmission. Another study of 32 men with COVID-19 revealed only one patient with SARS-CoV-2 in a semen sample [39]. The possibility that this rare finding was caused by contamination cannot be ruled out [42]. Saylem et al. evaluated the semen of 30 patients one day following their diagnosis with COVID-19 and were able to detect SARS-CoV-2 in four patients. In this study, all patients were advised to wash their hands and use masks prior to and during masturbation to avoid contamination [43]. To conclude, the evidence for the presence of SARS-CoV-2 in semen has been provided by case-control studies mainly, while no research has been conducted to assess the semen before and after SARS-CoV-2 infection.

The testes of patients with COVID-19 display seminiferous tubular injury, Sertoli cell swelling, reduced Leydig cell number and mild lymphocytic inflammation [44,45]. More specifically, in microscopic examination of the testes from 12 deceased COVID-19 patients, Sertoli cell swelling, vacuolation and cytoplasmic rarefaction and detachment from tubular basement membranes were observed. Moreover, in COVID-19 patients’ testes, there were lower Leydig cell numbers compared with deceased individuals without COVID-19 (2.2 vs. 7.8, *p* < 0.001). There was also edema and mild inflammation with T-lymphocyte and histiocyte infiltration in the interstitium. The presence of the virus was confirmed by RT-PCR in only 1 of 12 cases [46]. A small study showed several spermatogenesis abnormalities in 50% of autopsy specimens from men who died of COVID-19 [47]. Studies in humans have reported that infection with SARS-CoV-2 results in a deterioration of seminal parameters such as total sperm number (millions/ejaculation) and sperm concentration (millions/mL) [37]. Summarizing the evidence, it is more likely for SARS-CoV-2 exerts a direct effect on spermatogenesis rather than an interaction with sperm in semen.

The wide distribution of ACE2 receptors in the testes enable virus entry into the cells and its replication, together with ACE2 mRNA in spermatogonia, seminiferous duct cells, Leydig cells and Sertoli cells [48]. ACE2 receptors are differently distributed and regulated in men and women, with their expression being age-dependent, reaching the highest levels at around 30 years of age and the lowest at 60 years, in both sexes [49]. A new method was developed using a short synthetic peptide which targets ACE2 in human serum. This peptide connects to ACE2 from the serum and forms a protein complex that can be detected by electrochemical assays. In this way, the viral load of SARS-CoV-2 could be quantified, as the human body increases ACE2 serum concentrations to minimize the entry of the virus into the cells [50]. Therefore, as indicated by some recent clinical reports, serum ACE2 may serve as an index of viral load and its associated complications. Presence of ACE2 in the testes is more frequent in infertility, and infertile men are more susceptible to viral infection compared to fertile men [48,51].

A trial including 120 patients with a history of SARS-CoV-2 infection reported decreased semen parameters post-SARS-CoV-2 infection [52], while semen quality was restored 2 months after the infection. The above process is thought to be immunologically mediated, as increased anti-sperm IgA and IgG concentrations have been detected in the semen [52]. Another trial of 43 male patients with proven recovery from COVID-19 (age range 30–64 years) showed an increased risk of developing oligo- or cryptozoospermia [53]. Apart from COVID-19 per se, fever is a well-established risk factor that can impair sperm parameters. Fever leads to increased temperature in the testes that can deteriorate sperm quality. The detrimental effects of fever in the semen appear 74 days after the onset of fever (one spermatogenesis cycle) [54]. Commonly used medications (e.g., antibiotics, steroids) [55] in cases of SARS-CoV-2 infection may have a negative impact on male hormonal profiles and reproductive potential [56]. For example, steroids interfere with the hypothalamic–pituitary–gonadal axis [57].

More studies evaluating the semen of patients before and after the infection are needed to clarify the impact of SARS-CoV-2 on semen, with special focus on potential long-term effects and the reversibility of semen parameters [58]. Despite yet inconclusive data, it is being debated whether semen cryopreservation should be considered during the pandemic, given that SARS-CoV-2 infection is highly likely to lead to long-term detrimental effects on semen [59]. The Italian Society of Andrology and Sexual Medicine (SIAMS) suggests that patients recovered from COVID-19, especially those of reproductive age, should undergo a gonadal function evaluation, including a semen analysis [60].

Recently, a concern was expressed about the possible negative effect of COVID-19 vaccination on male fertility. Regarding the safety of COVID-19 vaccination, the two mRNA vaccines, BNT162b2 (Pfizer-BioNTech) and mRNA-1273 (Moderna), were studied regarding their impact on semen parameters. In one study, semen parameters were assessed before and after the COVID-19 vaccination in 45 healthy volunteers (age range 18–50 years) [61]. No decrease was observed in any semen parameter after two doses of COVID-19 mRNA vaccine. Similarly, another study including 43 men after vaccination with BNT162b2 showed no decrease in semen parameters [46]. It was suggested that the mRNA vaccine, which does not contain the live virus, would be unlikely to affect semen parameters. According to a statement from the Society for Male Reproduction and Urology in 2021, there is no definitive data to support this association [61]. A study designed to assess the semen parameters of infertile men demonstrated no impact of vaccinations on semen parameters or reproductive outcomes after assisted reproduction technology (ART) [62], regardless the vaccination type (mRNA vaccines or viral vector). Similar results were reported in studies involving healthy volunteers [63] or fertile men [64].

## 3. Possible Mediators of SARS-CoV-2 Impact on Male Infertility

### 3.1. Oxidative Stress

The pathophysiology of respiratory virus infections (including SARS-CoV-2) involves inflammation, cytokine production and cell death, with the latter associated with reactive oxygen species (ROS) overproduction and increased oxidative stress. ROS overproduction in SARS-CoV-2 triggers the nuclear factor kappa-light-chain-enhancer of the activated B-cell (NF-κB)-toll-like receptor pathway [65]. This mechanism further increases the release of cytokines, leading to inflammatory response exaggeration that may have a negative impact on male fertility [66]. Moreover, cytokines, such as interleukin-1β (IL-1β) and tumor necrosis factor-α (TNF-α), have been associated with reduced semen characteristics [67]. The major role of oxidative stress in inducing male infertility is illustrated by the term “Male Oxidative Stress Infertility (MOSI)”, to be used when male infertility is caused by oxidative stress [68].

Recently, autopsy-based studies provided evidence that SARS-CoV-2 infection is associated with increased oxidative stress leading to testicular cell apoptosis, while severe COVID-19 illness was associated with decreased interstitial tissue volume and seminiferous tubuli length [69]. According to a prospective cohort study, including 84 patients with COVID-19 and 105 healthy controls, the COVID-19 group had higher ROS concentrations in semen, which were over 1000 relative light units higher (*RLU*/s/10^6^ sperm) than the control group (400 RLU) throughout the study (*p* < 0.05). However, there was a decrease in ROS concentrations in the 60-day follow-up in the COVID-19 group (from >1000 to 700 RLU, *p* < 0.05); no change was observed in the control group [70]. Furthermore, ROS concentrations in semen samples diminished by 55.5% (*p* < 0.001) at 4 months after COVID-19 diagnosis compared with samples collected at the 14th day of infection [71].

The renin–angiotensin–aldosterone system (RAAS) plays a critical role in the pathogenesis of vascular disease and COVID-19. The SARS-CoV-2 enters the host cells through the ACE2 receptor cleavage by transmembrane protease, serine-2 (TMPRSS2). In addition, angiotensin II increase thrombin generation, possibly via a direct impact on tissue factors [72]. Interestingly, the testes have high concentrations of ACE2 mRNA and protein compared with other tissues in humans [73,74] (Figure 2). There is evidence that angiotensin II augments the production of ROS [72]. This mechanism leads to the generation of a proinflammatory phenotype in endothelial and vascular smooth muscle cells by the upregulation of adhesion molecules, chemokines and cytokines.

The severity of disease, quantified by leukocyte count and C-reactive protein (CRP) concentrations, does not correlate with the production of oxidant and antioxidant defenses (hydrogen peroxide, total antioxidant capacity, reduced oxidized glutathione) and oxidative damage (malondialdehyde, carbonyl, sulfhydryl). Severely ill patients with high serum leukocyte counts and CRP concentrations may not be a decisive factor for changes in the redox profile [75].

### 3.2. Metabolic Stress

SARS-CoV-2 infection can lead to increased metabolic stress, while, inversely, metabolic stress is associated with increased vulnerability to SARS-CoV-2 infection. SARS-CoV-2 infection results in earlier development or worsening of type 2 diabetes mellitus (T2DM) [76]. The infection is characterized by multiple immunological complications (e.g., redox storm, cytokine storm) that may lead to β-cell impairment and apoptosis, islet capillary rarefaction, hypoxia and abnormal remodeling [77]. In turn, T2DM may cause infertility and functional hypogonadism in males, along with sexual dysfunction, such as erectile and ejaculatory dysfunction, impaired semen parameters and delayed puberty. Similarly, obesity increases the risk of several disorders and diseases, including T2DM, sleep apnea, cardiovascular risk and thrombosis, contributing to increased metabolic stress. While obesity has been linked to abnormal semen parameters, its overall effect on hormone concentrations or sperm DNA integrity varies, probably due to different pathogenetic pathways involved in obesity and gonadal function [78]. Obesity, associated in the past with high risk for severe H1N1 influenza virus infection, is now regarded a risk factor for admission to intensive care units and mechanical ventilation in SARS-CoV-2-affected patients [79].

### 3.3. Psychological Stress

The prevalence of psychological stress and anxiety was increased even before the COVID-19 era [80]. SARS-CoV-2 infection is typically accompanied by severe psychological stress. A case–control study of 11,923,499 individuals in the UK verified that patients with a positive PCR test were presented with increased risk of psychiatric morbidity, fatigue and sleep problems in the following period [81]. A cross-sectional, single-centered study on infertile men coping with emotional reactions due to no access to fertility clinics due to COVID-19 revealed increased levels of stress, worry and frustration [81].

SARS-CoV-2 infection has a direct impact on cognitive function that can lead to “brain fog”, causing changes to the development and/or functioning of the central nervous system (CNS) [82]. This phenomenon is known as “silent” or “happy hypoxia” [83]. This is a form of hypoxia, usually without symptoms, which is due to a response to hyperventilation and dyspnea, which is inadequate for the increased CNS demands for oxygen. In this scenario, the impaired mitochondrial function fails to respond to the hypoxic microenvironment, and CNS changes are thus observed.

Psychological stress per se has been linked to several conditions and diseases, including hypertension, cardiovascular disease, obesity, depression, and mortality. In addition, it may adversely affect male fertility. Of note, psychological stress may lead to a cytokine storm that in turn can induce oxidative stress. In mouse models, chronic unpredictable stress induces disruptions in the blood–testis barrier and compromised semen characteristics [84]. Recently, male mice underwent chronic-restraint stress for 3 months in conical centrifuge tubes; during the stress period, mice were placed in boxes with attenuated sound and light. The application of stress to the father led to inherited reproductive disorders in the offspring, such as impaired semen parameters and infertility [85].

Psychological stress developed during the pandemic may have lead to a delay in attempting fertility. In an Italian study, one-third of infertile couples desiring fertility before the pandemic was found to have changed their decision during the lockdowns [86]. Precise numerical data on the overall impact of COVID-19 on fertility among various populations will not be available until some time has passed [87]. Anyway, quarantines and other restricting measures that were widely applied in several countries during the COVID-19 pandemic resulted in high rates of sexual dysfunction and reductions in sexual activity; the effect seems to be higher in women than men [88]. Sexual dysfunction could lead to male infertility. One out of six men of infertile couples has erectile dysfunction or premature ejaculation, and one out of ten has orgasmic dysfunction [89]. Furthermore, reduced intercourse frequency has a negative impact on couple fecundity [90]. It can be concluded that if a person’s psychological condition improves, the frequency of sexual intercourse is likely to increase, resulting in an increase in fertility.

## 4. Conclusions

There is increasing evidence of a possible adverse effect of SARS-CoV-2 infection on male fertility by simultaneously elevating metabolic, oxidative and psychological stress. If this hypothesis is confirmed, patients with underlying diseases, particularly those with T2DM and arterial hypertension, would be more likely to present with gonadal injury after SARS-CoV-2 and should be counseled accordingly. Most importantly, as RAAS is a key player in SARS-CoV-2 infection and its gonadal complications, an investigation into whether the use of RAAS blockers could result in a reduction in the SARS-CoV2 impact on male patients’ gonadal function would be of high clinical importance.

## Figures and Tables

**Figure 1 ijerph-19-05277-f001:**
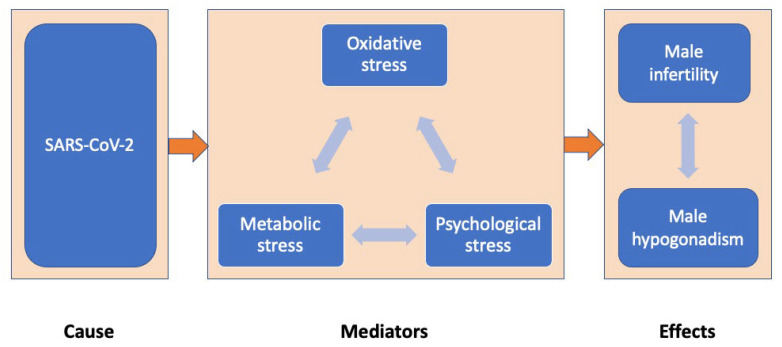
The impact of COVID-19 on male reproduction (via effects on testosterone secretion or semen parameters) through oxidative, metabolic and psychological stress mediators.

**Figure 2 ijerph-19-05277-f002:**
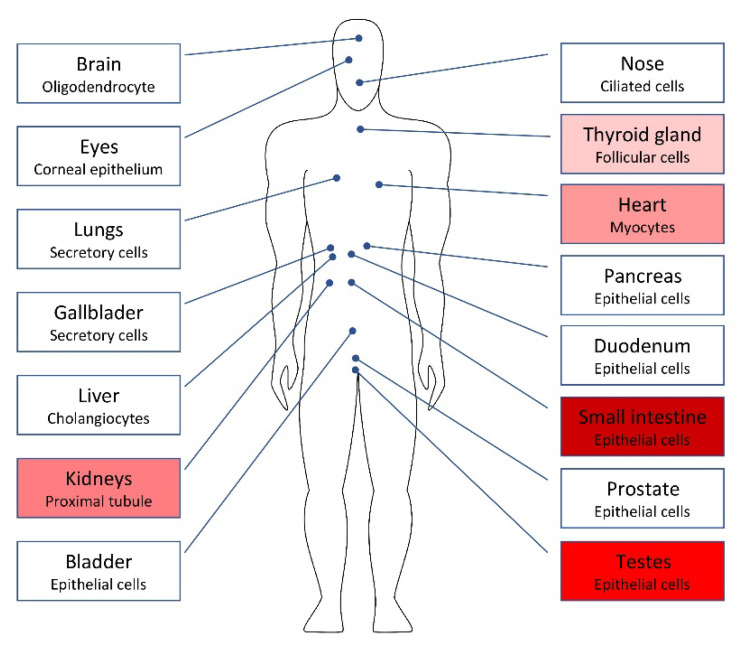
The distribution of angiotensin-converting enzyme-2 receptors (ACE2r) throughout the male body. Ιn red scale, the 5 organs (starting with small intestine and testes) with the widest distribution of ACE2r (Ref. [74]). A large presence of ACE2r is also found in adipose tissue, the breast and colon (not shown).

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
