# Peer review of "Metabolic, Oxidative and Psychological Stress as Mediators of the Effect of COVID-19 on Male Infertility: A Literature Review"

_ijerph, 2022, doi:10.3390/ijerph19095277_

Round 1
Reviewer 1 Report
This review needs extensive organization and editing to be fit for publication as follows:
Major comments:
1. A major concern in the current review is the lack of tables and figures highlighting the main results of the studies dealing with the topic or summarizing the mechanisms through which different stressors affect the progression of COVID-19. Moreover, Fig.1 is non-informative and does not add new to the topic. For instance, the authors could add a figure demonstrating the role of angiotensin-converting enzyme-2 receptors distributed in different organs of the body in the COVID-19 pathogenesis.
2. The way by which the authors cite references within paragraphs has many mistakes and needs to be carefully revised throughout the manuscript. For instance, on page 4 the second paragraph, ref 39 has been added in two consecutive paragraphs, page 5 ref 46 has been repeated three times in three consecutive paragraphs…etc. the previous error has been repeated many times throughout the manuscript.
3. The manuscript is full of typing errors (E.g. line 4 in the abstract womeb should be women), grammatical, and formatting errors that required extensive careful revision. Importantly, the manuscript needs to be carefully revised for the English language by native English speakers.
4. The use of abbreviations should be carefully revised throughout the manuscript. In the first line of the abstract E.g. Coronavirus Disease-2019 (COVID-19) then again in line 8.
5. Line numbering is highly recommended to be added to facilitate the reviewing process.
Other comments:
1. Keywords: remove “ Stress” and “metabolic stress; psychological stress”
2. Page 2: “There are several causes of infertility. Hypogonadotropic hypogonadism, hyperprolactinemia, infections, systemic diseases, cystic fibrosis, and in some cases lifestyle factors are causes that can affect males and females” the two sentences should be merged and rephrased and references should be added.
3. Page 2: “ T is an anabolic hormone secreted from testicles and the adrenal cortex”. The sentence should not begin with an abbreviation. Please, revise.
4. Page 3 line 1: change P < .0001 to P < 0.0001. Uniform throughout the manuscript.
5. Repetition of paragraphs with the same meaning exists in many sections. E.g. at the end of page 3 “Until now, no research has been performed to assess the semen of men before and after SARS-CoV-2” then again at the beginning of page 4.
6. Page 4: Human studies have reported that infection with SARS-CoV-2 results in a deterioration of sperm parameters [30]. Specify the types of sperm abnormalities in the earlier studies.
7. Page 6: “A risk factor for severe H1N1 influenza virus infection, obesity is now regarded as a risk factor for admission to critical care and invasive mechanical ventilation in SARS-CoV-2-affected patients”. Rephrase and add the reference for H1N1 influenza information.
8. Pages 6-7: “Βesides, it has recently been shown in mice that the application of stress to the father could lead to inherited reproductive disorders to the offspring such impaired sperm parameters and infertility”. Add the reference.
9. Page 7: “Recently, other mouse models have shown that the application of stress to the father could lead to inherited reproductive disorders to the offspring, such as impaired sperm parameters and infertility [67].” What is the type of stress applied to the father?
10. Page 7 “Therefore, further studies needed and also meta-analyses that will clarify the already existing data [71]” the review focus on male infertility. No need to add information on the effects on females. Delete it.
11. Conclusion: remove “ Therefore, further studies needed and also meta-analyses that will clarify the already existing data [71]”.
Reviewer 2 Report
Dear Authors,
This paper addresses an interesting topic, however, I would recommend several modifications before considering its publication. Below these are some suggestions for You:
Title:
- I suggest to underline in the title it is a literature review.
List of authors:
- Add dot after G in “Dimitrios G Goulis”
Review:
- Well organized and provides sufficient data.
Conclusions:
- Conclusions need to be revised. Do not make a summarize of the review, but make conclusions - only suggest the clinical significance of the outcome of your findings
Author contributions:
- You have forgotten to add them in the manuscript. You have to add them according to Instructions for authors.
References:
- The article is well documented with a vast number of references.
Best regards and good luck
Reviewer 3 Report
Summary:
This review examines the role of SARS-CoV-2 in male infertility via the sperm oxidative stress hypothesis, as well as the impacts of metabolic stress and psychological stress. This review covers a range of topics that relate to these topics, including the changes in the hormone profiles, semen parameters, testicular pathology, and aspects of sperm health. While this reviewer believes it should be made clear the oxidative stress mediator is a hypothesis, albeit a strong one, this review does provide a good summary of the compelling evidence in its support. As the different types of stress are the driving force behind the review, there needs to be stronger links made throughout the document to bind the different topics raised here together, i.e. how all these topics link together to form a cohesive mechanism. This is briefly touch on in the conclusion, but should be given priority throughout. The immunological mechanisms underlying many topics discussed should also be more detailed. The document also requires some proof-reading.
Abstract:
Line 4: Typo, ‘womeb’ should be ‘women’?
Line 3-5: There seems to be a big jump between ‘men present higher severity and fatality’ and this being a reproductive tract disease. Please add information to link these two ideas or re-order the information in the abstract to link the ideas. Men presenting with worse disease could be due to many factors, physiological or otherwise, other than the reproductive tract as you outline later in the abstract.
Line 8: No need to define COVID-19 here, you’ve already defined it in line 1.
Line 12: Suggest changes ‘parameters’ to ‘health’. … ’a decline of sperm health.’ 1.
Introduction:
Line 3-4: Suggest outright saying SARS-CoV-2 is the causative agent of COVID-19, for clarity.
Line 15: Please add additional references to the male disease bias statement. As this is the foundation for the review, there should be more than one reference. Within the cited paper (Reference 8), there are some additional resources that can also be cited, which are broader than R8, which is research focussed on age stratification.
Line 16: Please provide references for the reproductive hormone information, or reword to make it clear this is a hypothesis. Consider adding ‘for example’ to the list of causes for infertility, or otherwise add more information like the range of genetic conditions, Klinefelter’s, Sertoli cell only syndrome, germ cell arrest, etc. The authors could consider adding a short description of the testicular structure too, since it becomes important later in the review of testicular histopathology. It seems to be an oversimplification that male infertility, and male infertility caused by SARS-Cov-2, can be narrowed down to oxidative stress, metabolic stress, and psychological stress. This hypothesis ignores the myriad of other factors, like immune responses and autoimmune orchitis for example, which come before and/or act in conjunction with oxidative stress or even without oxidative stress to damage sperm. Please include this sort of information in this manuscript.
https://onlinelibrary.wiley.com/doi/10.1111/andr.13098 here’s a recent short review on this topic for reference. 2. SARS-CoV-2 and male infertility 2.1 SARS-CoV-2 and testosterone A nice summary of current knowledge presented in a logical flow. T is given plenty of attention, but DHT is only mentioned in the opening sentence. Can DHT be given more attention here too? 2.2 SARS-CoV-2 and semen Reference needed for the 20 viruses found in human semen information. Where reference 26 is in the 4th line, it would be better to cite the original research rather than a review. This principle should be applied throughout the document if needed. Suggest placing reference 28 after both applicable sentences, rather than just the second sentence, for clarity. “On the other hand, the testes of patients with COVID-19 display seminiferous tubular injury, Sertoli cell swelling, reduced Leydig cell number, and mild lymphocytic inflammation. More specifically, in microscopically examination of testis from 12 deceased COVID-19 patients was observed swelling of sertoli cells, vacuolation and cytoplasmic rarefaction and detachment from tubular basement membranes. Moreover, in COVID-19 patients’ testes was observed significantly lower number of Leydig cells compared with deceased individuals without COVID-19 (2.2 vs 7.8, p < 0.001). There was also edema and mild inflammation with T-lymphocytes and histiocytes infiltration in the interstitium. The presence of the virus certified by Reverse transcription Polymerase chain reaction (RT-PCR) in only one of 12 cases [33].” This part of the paragraph requires more referencing for the large amount of information. It would be good to have more than one reference/study to back up this pretty important information, too. There seems to be some double up with the following few sentences, so perhaps this information could be combined here. “Human studies have reported that infection with SARS-CoV-2 results in a deterioration of sperm parameters [30], the latter likely on account of the wide distribution of ACE2 receptors, which enable virus entry into the cell and its replication in the testes, together with ACE2 mRNA in spermatogonia, seminiferous duct cells, as well Leydig, and Sertoli cells[26].” Suggest that the transition into ACE2 information needs to be a separate paragraph in this section. This sentence also needs rewriting for clarity. The switching between cells and tissues makes it unclear what you are suggesting supports viral replication, the sperm itself, or the cells supporting spermatogenesis. “ACE2 receptors are differently distributed and regulated in men and women, their expression being age-dependent and reaching the highest levels at around 30 years and the lowest at 60 years of age in both sexes. Recently, a new method was developed using a short synthetic peptide which targets the ACE2 in human serum. In particular, this peptide connects to ACE2 from the serum, and then form a protein complex that is detected by electrochemical assays. In this way, the viral load of SARSCoV-2 of affected patients could be predicted, as the human body increases ACE2 serum concentrations in order to minimize the entry of the virus into the cells through cell membrane [36].” This part of the paragraph needs more references. Please apply this principle throughout the document as needed. “Moreover, cytokines, such as interleukin-1β (IL-1β) and tumor necrosis factor-α (TNF-α) have been associated with reduced sperm characteristics [38].” This information needs to be contextualised. It’s placed at the end of a lot of information about ACE2, and seems important, but there is nothing to link how these cytokines relate to ACE2 expression. If they do not relate to ACE2 expression, this information should be somewhere else, possibly in the testicular histopathology section as pro-inflammatory cytokine production in the testis is detrimental and may contribute to pathology. “A large trial including 120 patients with a history of COVID-19 reported infection decreased semen parameters post COVID infection[39]” Switched here to COVID infection rather than SARS-CoV-2 infection, please amend. Further down in the paragraph, there is also Covid-19 in lowercase, please amend. “(i.e. antibiotics, steroids)” A reference is required for the impact of antibiotics on sperm health. 3.1 Oxidative stress “The pathophysiology of respiratory virus infections (including SARS-CoV-2) involves inflammation, cytokine production, and cell death, the latter associated with reactive oxygen species (ROS) overproduction and increased oxidative stress ROS overproduction in SARS-CoV-2 triggers the nuclear factor kappa-light chain-enhancer of the activated B cell (NF-κB)-toll-like receptor pathway[50], this possibly having a negative impact on male fertility[51].” This is a long and confusing sentence. Suggest breaking into two sentences and rewording for clarity. It’s not clear how respiratory viruses involving inflammation skip right through to male infertility. The middle parts of this process need to be clear, you could just be talking about lung inflammation here. OS needs to be defined on first use. Are reductions in OS/ROS at the 60 days or 4 months post infection due to completion in spermatogenesis cycles as you previous described? Might be good to link these concepts, if this is what you believe. Or possibly give an alternative hypothesis. Please refine the paragraph on ACE2 in this section to avoid repetitive information and contextualise the relevant information for either the semen or more likely the testicular environment. 3.2 Metabolic stress Suggested that as Metabolic Stress in in the title of the manuscript, its contribution needs to be more detailed. E.g. the direct effect on the testis, semen, and sperm could be given more attention. 3.3 Psychological stress Suggested that you specify the pre-COVID-19 period here, rather than just the pre-COVID period. Unless you might have some information on the other coronavirus disease periods to mention here to support the hypothesis? Please specify what impact the high rates of sexual dysfunction and reduced sexual activity has on male fertility. If people were to recover psychologically, would they still be infertile? This may undermine the proposed mechanism. 4. Conclusions A fair summary of the proposed mechanism.
Round 2
Reviewer 1 Report
No further comments to be addressed
Reviewer 3 Report
Thank you for making a great many changes to improve this manuscript. There are still a small number of changes that are required.
Line 70: Leydig cells are not the only cell type in the interstitial compartment of the testes, there are also testicular macrophages and a small population of other immune cells. Also suggest to mention the pathway that sperm takes to reach the semen, from the testes onward. Please amend.
Line 130: genetic should be genital?
Line 168: This information seems out of place. Do you mean to say that, SARS-CoV-2 is infrequently found in semen but is frequently found in testes, indicating a direct effect on spermatogenesis, rather than interaction with sperm in semen? Please give this information some context.
Figure 2: It would be more impactful to indicate, along with the distribution, the abundance of ACE-2, which would give support for your hypothesis with high abundance in the male reproductive tract.
Author Response
Please see attachment.

This manuscript is a resubmission of an earlier submission. The following is a list of the peer review reports and author responses from that submission.